# Tranilast directly targets NLRP3 to treat inflammasome-driven diseases

Yi Huang[1,2,†], Hua Jiang[1,†], Yun Chen[3], Xiaqiong Wang[1], Yanqing Yang[4], Jinhui Tao[5], Xianming Deng[3], Gaolin Liang[6], Huafeng Zhang[1,2] (ID), Wei Jiang[1,*] (ID) & Rongbin Zhou[1,2,**] (ID)

## Abstract

The dysregulation of NLRP3 inflammasome can cause uncontrolled inflammation and drive the development of a wide variety of human diseases, but the medications targeting NLRP3 inflammasome are not available in clinic. Here, we show that tranilast (TR), an old anti-allergic clinical drug, is a direct NLRP3 inhibitor. TR inhibits NLRP3 inflammasome activation in macrophages, but has no effects on AIM2 or NLRC4 inflammasome activation. Mechanismly, TR directly binds to the NACHT domain of NLRP3 and suppresses the assembly of NLRP3 inflammasome by blocking NLRP3 oligomerization. *In vivo* experiments show that TR has remarkable preventive or therapeutic effects on the mouse models of NLRP3 inflammasome-related human diseases, including gouty arthritis, cryopyrin-associated autoinflammatory syndromes, and type 2 diabetes. Furthermore, TR is active *ex vivo* for synovial fluid mononuclear cells from patients with gout. Thus, our study identifies the old drug TR as a direct NLRP3 inhibitor and provides a potentially practical pharmacological approach for treating NLRP3-driven diseases.

**Keywords** directly bind; inflammasome-driven diseases; inhibitor; NLRP3; tranilast

**Subject Categories** Immunology; Pharmacology & Drug Discovery

## Introduction

NLRP3 inflammasome is a protein complex formed by NOD-like receptor (NLR) family member NLRP3, adaptor protein ASC, and caspase-1 (Martinon *et al*, 2009; Davis *et al*, 2011; Jo *et al*, 2016). A variety of factors derived from not only pathogen, but also environment or host, can activate NLRP3 inflammasome to result in pyroptosis and the release of secretion of several proinflammatory cytokines, such as IL-1β or IL-18 (Chen *et al*, 2009). The dysregulation of NLRP3 inflammasome has been reported to be involved in the pathogenesis of several human diseases. Mutations in *NLRP3* gene can result in spontaneous NLRP3 inflammasome activation and are associated with cryopyrin-associated autoinflammatory syndromes (CAPS), which are a group of rare, inherited, autoinflammatory diseases (Broderick *et al*, 2015). In addition, NLRP3 inflammasome can sense some host-derived "danger signals", including monosodium urate crystals (MSU), cholesterol crystals, amyloid-β aggregates, unsaturated fatty acids, high glucose, and ceramide, to promote chronic inflammation and contribute to the development of human complex diseases, including gout, atherosclerosis, neurodegenerative diseases, and type 2 diabetes (T2D) (Martinon *et al*, 2006; Duewell *et al*, 2010; Masters *et al*, 2010; Zhou *et al*, 2010; Wen *et al*, 2011; Heneka *et al*, 2012; Lamkanfi & Dixit, 2012). Thus, NLPR3 inflammasome has been regarded as a potential drug target for the treatment of inflammatory diseases.

In recent years, a few small-molecule compounds have shown potential inhibitory effects on NLRP3 inflammasome activation *in vitro*, including dimethyl sulfoxide (DMSO), 3,4-methylenedioxy-β-nitrostyrene, glyburide, parthenolide, sulforaphane, isoliquiritigenin, MCC950, β-hydroxybutyrate, flufenamic acid, and mefenamic acid (Lamkanfi *et al*, 2009; Juliana *et al*, 2010; Yan *et al*, 2013; Ahn *et al*, 2014; He *et al*, 2014; Honda *et al*, 2014; Coll *et al*, 2015; Youm *et al*, 2015; Daniels *et al*, 2016; Yang *et al*, 2016). Among them, several compounds have been tested in animal models of human diseases. Sulforaphane and isoliquiritigenin have been shown to alleviate high-fat diet (HFD)-induced metabolic disorders in mice (Honda *et al*, 2014; Yang *et al*, 2016). MCC950 and β-hydroxybutyrate have shown beneficial effects in mice models of CAPS (Coll *et al*, 2015; Youm *et al*, 2015). MCC950, flufenamic acid, and mefenamic acid have been shown to delay the development of AD in

1   Institute of Immunology and the CAS Key Laboratory of Innate Immunity and Chronic Disease, CAS Center for Excellence in Molecular Cell Sciences, Hefei National Laboratory for Physical Sciences at Microscale, University of Science and Technology of China, Hefei, China
2   Innovation Center for Cell Signaling Network, University of Science and Technology of China, Hefei, China
3   State Key Laboratory of Cellular Stress Biology, Innovation Center for Cell Signaling Network, School of Life Sciences, Xiamen University, Xiamen, Fujian, China
4   Department of Clinical Laboratory, The First Affiliated Hospital of Bengbu Medical College, Bengbu, China
5   Department of Rheumatology & Immunology, The First Affiliated Hospital of University of Science and Technology of China, Hefei, Anhui, China
6   CAS Key Laboratory of Soft Matter Chemistry, Department of Chemistry, University of Science and Technology of China, Hefei, China
    *Corresponding author. Tel: +86-551-63600125; E-mail: ustcjw@ustc.edu.cn
    **Corresponding author. Tel: +86-551-63600302; Fax: +86-551-63600831; E-mail: zrb1980@ustc.edu.cn
    †These authors contributed equally to this work

mice model (Daniels *et al*, 2016; Dempsey *et al*, 2017). Although the beneficial effects of these compounds on NLRP3-driven diseases in animal models are promising, none of them have been tested in clinic. Thus, it is urgent to develop NLRP3 inflammasome inhibitors with high safety for clinical trials.

Tranilast (N-[3′,4′-dimethoxycinnamoyl]-anthranilic acid, TR) is an analog of a tryptophan metabolite and has been reported to have an inhibitory effect on homologous passive cutaneous anaphylaxis because it can inhibit IgE-induced histamine release from mast cells (Azuma *et al*, 1976; Koda *et al*, 1976; Platten *et al*, 2005). Subsequently, TR has been clinically used in the treatment of a variety of inflammatory diseases, including bronchial asthma, atypical dermatitis, allergic conjunctivitis, and hypertrophic scars (Darakhshan & Pour, 2015). Moreover, TR is a relatively safe drug and is well tolerated by most patients, at doses of up to 600 mg/day over a period of months (Darakhshan & Pour, 2015). Although TR has shown favorable pharmacological effects against inflammatory diseases, the molecular mechanisms and the direct target of its anti-inflammatory activity are still unknown.

In this study, we showed that TR directly bound to NLRP3 and inhibited NLRP3 inflammasome assembly and the subsequent caspase-1 activation and IL-1β production. More importantly, TR could prevent or treat NLRP3-dependent inflammatory diseases in mice models and was also active *ex vivo* for samples from patients with gout.

# Results

### TR specifically inhibits NLRP3 inflammasome activation in macrophages

To confirm the inhibitory effects of TR on inflammasome activation, we first examined whether TR inhibited caspase-1 cleavage and IL-1β secretion. We indeed observed that TR treatment blocked nigericin-induced caspase-1 cleavage, IL-1β secretion, and pyroptosis (Fig 1A and B, and Appendix Fig S1A). It has been reported that TR can inhibit cytokine-induced NF-κB activation (Oh *et al*, 2010), we then examined whether TR inhibited NLRP3 inflammasome activation via regulating the expression of NF-κB-dependent NLRP3 or pro-IL-1β expression. When BMDMs were stimulated with TR for 30 min after 3-h LPS treatment, TR had no effect on LPS-induced NLRP3, pro-IL-1β expression, TNF-α, or IL-6 production (Fig 1C and D, and Appendix Fig S1B–D), suggesting that TR-induced NLRP3 inflammasome inhibition was not caused by the downregulation of NLRP3 or pro-IL-1β expression at this condition. In contrast, when BMDMs were stimulated with TR for 30 min before 3-h LPS treatment, TR inhibited LPS-induced pro-IL-1β expression and IL-6 production, but had minimal effects on NLRP3 expression and TNF-α production (Appendix Fig S1B–D). These results suggest that TR can suppress both NLRP3 inflammasome activation and pro-IL-1β expression. In order to clarify the mechanisms underlying TR-induced inflammasome inhibition, we stimulated BMDMs with TR after 3-h LPS treatment in the later experiments. The observed inhibitory effects of TR on IL-1β secretion were also confirmed in human THP-1 macrophages (Fig 1E). Taken together, these results indicate that TR has the potential to inhibit caspase-1 activation and IL-1β secretion.

In addition to nigericin, NLRP3 inflammasome can be activated by both pathogen-associated molecular patterns (PAMPs) and danger-associated molecular patterns (DAMPs), including aluminum salts (Alum), ATP, monosodium urate crystals (MSU), and cytosolic LPS (cLPS) (Davis *et al*, 2011; Kayagaki *et al*, 2011). To determine whether TR was a common inhibitor for NLPR3 inflammasome, we examined other NLRP3 agonists. Pretreatment with TR inhibited caspase-1 cleavage and IL-1β secretion triggered by all examined agonists, including MSU, Alum, ATP, and cLPS (Fig 1F–I), similar to nigericin. Interestingly, TR could not block cLPS-induced gasdermin D (Gsdmd) activation and pyroptosis, suggesting that TR targets the downstream of caspase-11 to inhibit non-canonical NLRP3 activation (Appendix Fig S1E and F). These results indicate that TR is a potent and broad inhibitor for the activation of both canonical and non-canonical NLRP3 inflammasome activation. We also tested whether TR could inhibit other inflammasomes, such as NLRC4 and AIM2 inflammasome. The results showed that TR had no effect on NLRC4 or AIM2 inflammasome activation, which were triggered by *Salmonella typhimurium* (*Salmonella*) infection or poly A:T transfection, respectively (Appendix Fig S2A and B). Taken together, these results demonstrate that TR can specifically inhibit NLRP3 inflammasome activation and the subsequent IL-1β production.

### TR has no effects on upstream signaling of NLRP3, but inhibits the assembly of NLRP3 inflammasome

We then investigated how TR inhibited NLRP3 inflammasome activation. Although TR has been reported to exert different biological effects, the targets of TR have not been validated. TR has been proposed to inhibit transient receptor potential cation channel subfamily V member 2 (TRPV2) (Zhang *et al*, 2012), and we then examined whether TR blocked NLRP3 inflammasome *via* inhibition of TRPV2. The results showed that knockdown of TRPV2 in BMDMs had no effect on nigericin-induced NLRP3 inflammasome activation (Appendix Fig S3A and B), suggesting TRPV2 is not involved in TR-induced NLRP3 inflammasome inhibition. Previous study has shown that TR can inhibit the activity of hematopoietic prostaglandin D2 synthase (HPGDS) (Ikai *et al*, 1989), but our results showed that HPGDS was not involved in NLRP3 inflammasome activation (Appendix Fig S3C and D).

During NLRP3 inflammasome activation, ASC oligomerization is a critical step for the subsequent caspase-1 activation (Lu *et al*, 2014; Dick *et al*, 2016). Consistent with the inhibitory effects of TR on caspase-1 activation and IL-1β production, TR remarkably suppressed nigericin-induced ASC oligomerization (Fig 2A). These results suggest that TR acts upstream of caspase-1 and ASC oligomerization to inhibit NLRP3 inflammasome activation. We then studied whether TR affected potassium efflux, which is an upstream signaling event of NLRP3 activation (Petrilli *et al*, 2007; Munoz-Planillo *et al*, 2013). Nigericin treatment could result in dramatically decrease in intracellular potassium, but this effect was not suppressed by TR (Appendix Fig S4A), suggesting that TR has no effect on potassium efflux during NLRP3 inflammasome activation. Mitochondrial damage, represented as mitochondria fission, clustering, and ROS production, is another upstream signaling event of NLRP3 activation (Zhou *et al*, 2011). Nigericin-induced mitochondrial damage and

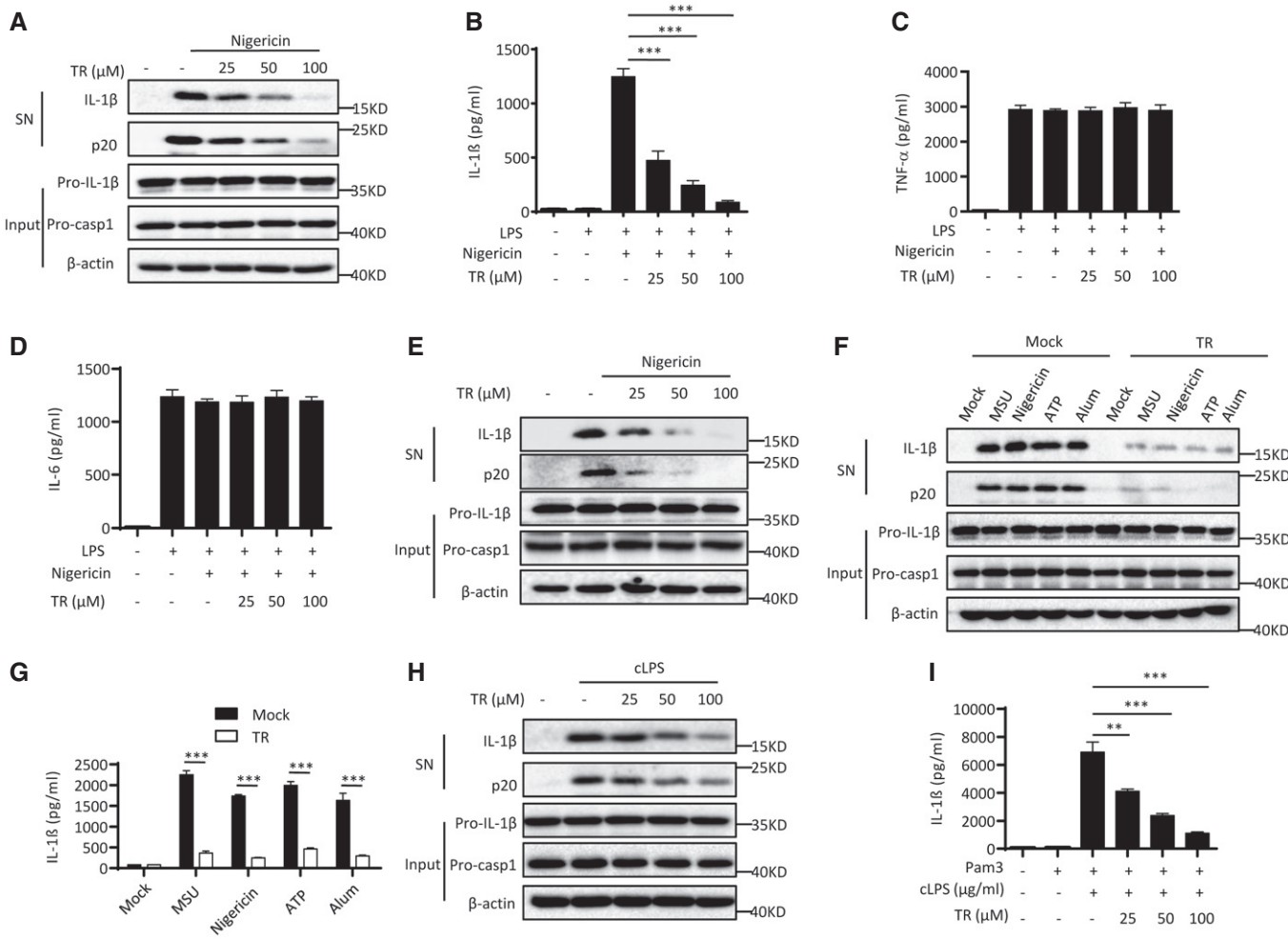

**Figure 1. TR specifically inhibits NLRP3 inflammasome activation.**

A Immunoblot analysis of IL-1β and cleaved caspase-1 (p20) in culture supernatants (SN) of LPS-primed BMDMs treated with various doses (above lanes) of TR and then stimulated with nigericin, and immunoblot analysis of the precursors of IL-1β (pro-IL-1β) and caspase-1 (pro-casp1) in lysates of those cells (Input).

B–D ELISA of IL-1β (B), TNF-α (C) and IL-6 (D) in supernatants from LPS-primed BMDMs treated with various doses (above lanes) of TR and then stimulated with nigericin.

E Immunoblot analysis of IL-1β and cleaved caspase-1 (p20) in supernatants from PMA-differentiated THP-1 cells treated with various doses (above lanes) of TR and then stimulated with nigericin.

F Immunoblot analysis of IL-1β and cleaved caspase-1 (p20) in culture supernatants (SN) of LPS-primed BMDMs treated with TR (100 μM) and then stimulated with MSU, nigericin, ATP, Alum, and immunoblot analysis of the precursors of IL-1β (pro-IL-1β) and caspase-1 (pro-casp1) in lysates of those cells (Input).

G ELISA of IL-1β in culture supernatants (SN) of LPS-primed BMDMs treated with TR (100 μM) and then stimulated with MSU, nigericin, ATP, Alum.

H Immunoblot analysis of IL-1β and cleaved caspase-1 (p20) in culture supernatants (SN) of Pam3-primed BMDMs treated with various doses (above lanes) of TR and then stimulated with cytosolic LPS (cLPS), and immunoblot analysis of the precursors of IL-1β (pro-IL-1β) and caspase-1 (pro-casp1) in lysates of those cells (Input).

I ELISA of IL-1β in culture supernatants (SN) of Pam3-primed BMDMs treated with various doses (above lanes) of TR and then stimulated with cytosolic LPS (cLPS).

Data information: Data are from three independent experiments with biological duplicates in each (B–D, G, I; mean and s.e.m. of $n = 6$) or are representative of at least three independent experiments (A, E, F, H). Statistics were analyzed using an unpaired Student's t-test: **$P < 0.01$, ***$P < 0.001$.

Source data are available online for this figure.

ROS production were normal in BMDMs pretreated with TR (Appendix Fig S4B), indicating TR does not affect mitochondrial damage in NLRP3 inflammasome activation. In addition, chloride efflux has been proposed as another upstream signaling event for NLRP3 activation (Daniels *et al*, 2016; Tang *et al*, 2017), but TR could not block nigericin-induced decrease in intracellular chloride in BMDMs (Appendix Fig S4C), suggesting that TR has no effect on the chloride efflux upstream of NLRP3 activation. Thus, these results suggest that TR acts downstream of potassium efflux, mitochondrial damage, and chloride efflux to inhibit NLRP3 activation.

Next, we assessed whether TR inhibited the assembly of NLRP3 inflammasome. Recently, NEK7 has been proposed as an essential component of NLRP3 inflammasome and NEK7–NLRP3 interaction is important for NLRP3 oligomerization and inflammasome assembly (He *et al*, 2016; Schmid-Burgk *et al*, 2016; Shi *et al*, 2016). Consistent with these previous studies, nigericin treatment promoted the endogenous interaction between NEK7 and NLRP3,

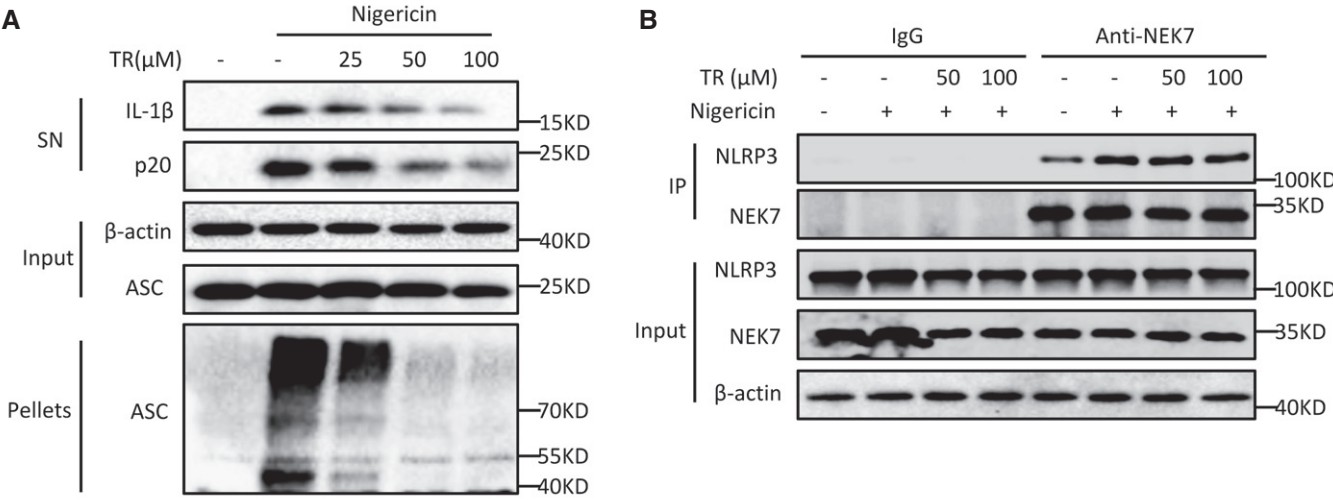

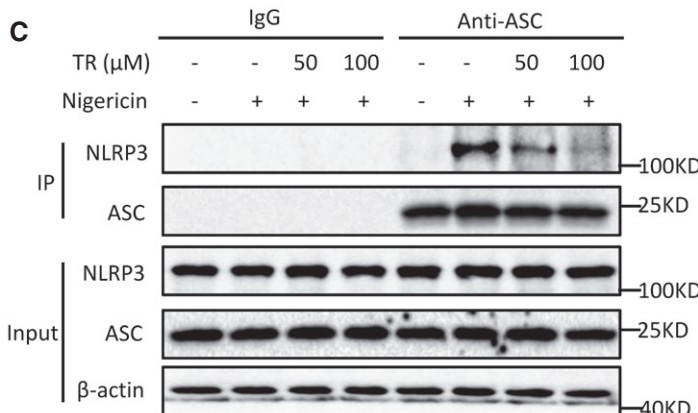

**Figure 2. TR suppresses the assembly of NLRP3 inflammasome.**

A   Immunoblot analysis of ASC oligomerization in lysates of BMDMs treated with various doses (above lanes) of TR for 30 min and then stimulated with nigericin for another 30 min.

B   Immunoprecipitation (IP) and immunoblot analysis of the interaction of endogenous NLRP3 and NEK7 in LPS-primed BMDMs treated with various doses (above lanes) of TR for 30 min and then stimulated with nigericin for another 20 min.

C   IP and immunoblot analysis of the interaction of endogenous NLRP3 and ASC in LPS-primed BMDMs treated with various doses (above lanes) of TR for 30 min and then stimulated with nigericin for another 20 min.

Data information: Data are representative of two or three independent experiments.
Source data are available online for this figure.

which could not be suppressed by TR treatment (Fig 2B). Another critical step for NLRP3 inflammasome assembly is the recruitment of ASC to NLRP3 (Martinon *et al*, 2009; Davis *et al*, 2011). We next determined whether TR could impact on NLRP3–ASC interaction and found that pretreatment with TR inhibited the endogenous interaction between NLRP3 and ASC in nigericin-treated macrophages (Fig 2C). Thus, these results indicate that TR blocks NLRP3-ASC complex formation to inhibit NLRP3 inflammasome activation.

**TR directly binds to NLRP3 and inhibits its oligomerization**

Since TR inhibited NLRP3-ASC complex formation, we next investigated whether TR bound to these proteins. A synthesized

biotinylated analog of TR (biotin-TR) was used as an affinity reagent and incubated with cell lysates of LPS-primed BMDMs or PMA-differentiated THP-1 cells. Compound was pulled down with streptavidin beads, and NLRP3 component proteins were detected by immunoblot analysis. The data showed that NLRP3, but not ASC or NEK-7, was pulled down by biotin-TR (Fig 3A and B). To determine whether TR interacts with NLRP3 directly, purified recombinant NLRP3 protein could be pulled down with biotin-TR (Fig 3C and Appendix Fig S5A), confirming that TR directly interacts with NLRP3. We next studied whether TR bound to other innate immune sensors. Flag-tagged NLRP3, NOD1, NOD2, NLRP1, or NLRC4 were overexpressed in HEK-293T cells, and the cell lysates were then incubated with biotin-TR. The results showed that only NLRP3

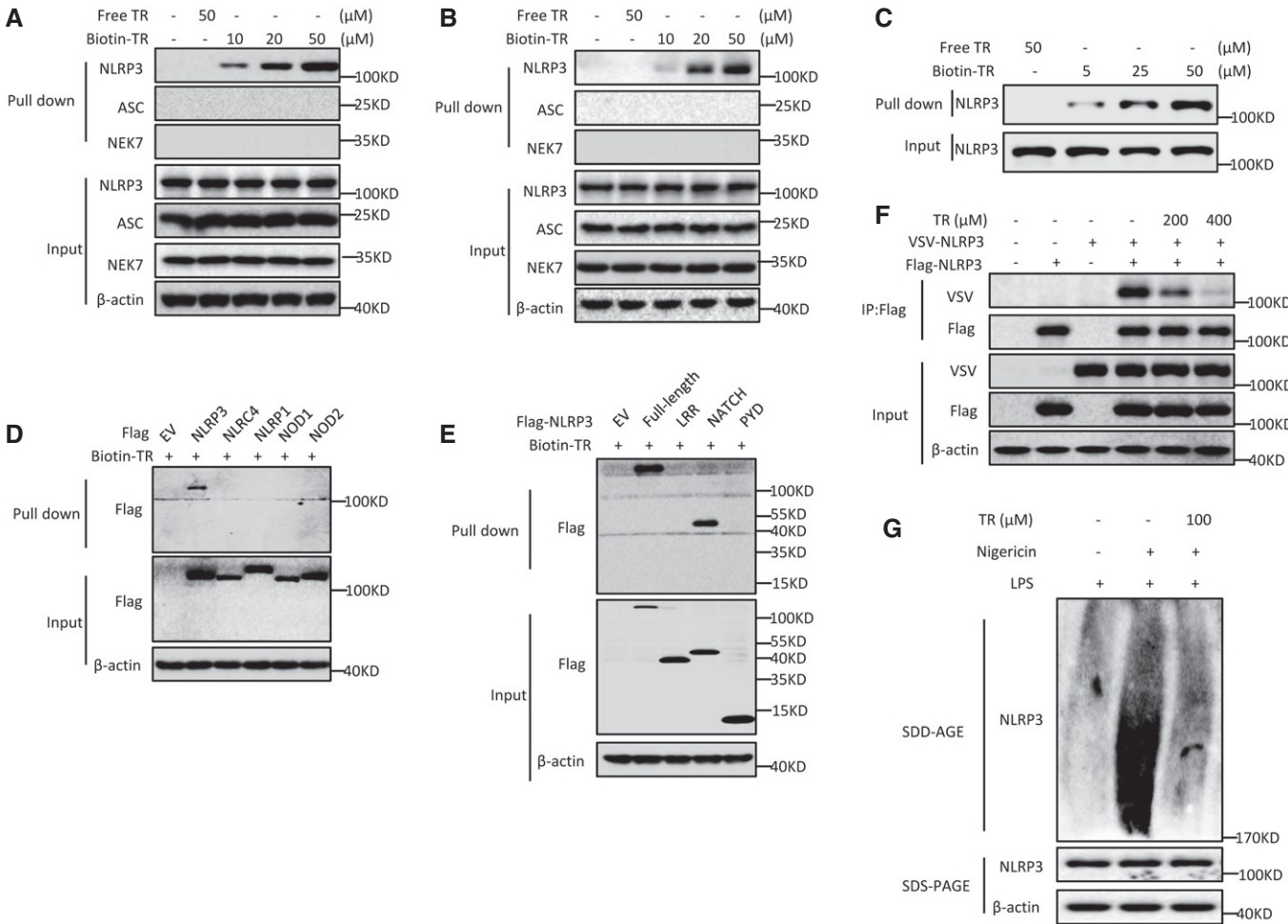

**Figure 3. TR directly binds to NLRP3 and inhibits its oligomerization.**

A, B   Cell lysates of LPS-primed BMDMs (A) or PMA-differentiated THP-1. (B) Cells were incubated with different concentrations of biotin-TR for 1 h, which were then pulled down with streptavidin beads.

C   Purified recombinant NLRP3 protein was incubated with different concentrations of biotin-TR for 1 h, which were then pulled down with streptavidin beads.

D, E   Cell lysates from HEK-293T cells transfected with Flag-tagged NLRP3, NOD1, NOD2, NLRP1, NLRC4, NLRP3-LRR, NLPR3-NACHT, or NLRP3-PYD constructs were incubated with different concentrations of biotin-TR for 1 h, which were then pulled down with streptavidin beads.

F   Immunoprecipitation (IP) and immunoblot analysis of the interaction of Flag-NLRP3 and VSV-NLRP3 in the lysates of HEK-293T cells. TR was added at 8 h post-transfection.

G   Immunoblot analysis of NLRP3 by SDD-AGE or SDS–PAGE assay in BMDMs treated with TR for 30 min and then left stimulated with nigericin for 20 min.

Data information: Data are representative of two or three independent experiments.
Source data are available online for this figure.

could be pulled down (Fig 3D), suggesting that TR specifically binds with NLRP3. NLRP3 contains three functional domains, LRR, NACHT, and PYD. We then studied which domain was responsible for the binding between NLRP3 and TR, and the results showed that only NACHT domain of NLRP3 bound TR (Fig 3E). These results indicate that TR directly binds to the NACHT domain of NLRP3.

We then investigated how the binding of TR to NACHT inhibited the assembly of NLRP3 inflammasome. The NACHT domain is important for NLRP3 oligomerization, which is a critical step for the assembly of NLRP3 inflammasome (Martinon *et al*, 2009; Davis *et al*, 2011). We then investigated whether TR could prevent the direct NLRP3–NLRP3 interaction. HEK-293T cells were transfected with Flag-NLRP3 and VSV-NLRP3 and treated with TR, and then, a

co-immunoprecipitation assay was performed. The results showed that the direct NLRP3–NLRP3 interaction was also suppressed by TR treatment (Fig 3F). In contrast, the direct interaction between NLRP3 and ASC was not altered (Appendix Fig S5B). The effect of TR on the endogenous NLRP3 oligomerization was confirmed using semi-denaturing detergent agarose gel electrophoresis (SDD–AGE) (Fig 3G), a method for detecting large protein oligomers in studying prions (Alberti *et al*, 2009; Hou *et al*, 2011). Since the ATPase activity of NACHT domain of NLRP3 is critical for NLRP3 oligomerization (Duncan *et al*, 2007), we then evaluated whether TR inhibited the ATPase of recombinant NLRP3 protein and found that TR had no effect on NLRP3 ATPase activity (Appendix Fig S5C), suggesting that TR blocks NLRP3 oligomerization *via* an ATPase-independent

manner. Thus, these results indicate TR can directly bind to the NACHT domain of NLRP3 and inhibit NLRP3 oligomerization by blocking direct NLRP3–NLRP3 interaction.

### TR inhibits NLRP3 activation *in vivo* and has beneficial effects in mouse models of gouty arthritis and CAPS

Since TR inhibits NLRP3 inflammasome activation *in vitro*, we then determined to analyze the activity of TR *in vivo*. Intraperitoneal injection of MSU elicited an NLRP3-dependent peritonitis characterized by IL-1β production and massive neutrophil influx (Martinon *et al*, 2006). Consistent with the role of TR in macrophages, TR treatment *in vivo* efficiently suppressed MSU-induced IL-1β production and neutrophil influx (Fig 4A and B). We also compared the anti-inflammasome activity of TR with MCC950, which is a selective inhibitor for NLRP3 inflammasome (Coll *et al*, 2015). Although the *in vitro* inhibitory activity of TR on MSU-induced IL-1β secretion was around 400 times less potent than MCC950 (Appendix Fig S6A), its *in vivo* activity on MSU-induced peritonitis was only around 5–10 times less potent than MCC950 (Appendix Fig S6B and C). The deposition of MSU is the major cause for the development of arthritis in the patients with gout (McQueen *et al*, 2012). Delivery of MSU to the joints of mice leads to NLRP3 inflammasome-dependent inflammation and pathology (Reber *et al*, 2014). Consistent with this, MSU injection induced acute joint swelling, which could be alleviated by *Nlrp3* deficiency or oral TR treatment (Fig 4C). TR also suppressed MSU-induced NLRP3-dependent IL-1β production in joint tissue (Fig 4D). Thus, these results indicate that TR is active *in vivo* and can prevent NLRP3-dependent acute inflammation and tissue damage.

NLRP3 gain-function mutation has been shown to cause Muckle–Wells syndrome (MWS), which is an autoinflammatory syndrome characterized by an excessive secretion of IL-1β (Ting *et al*, 2006). NLRP3-mutant mice with specific expression of MWS-associated mutation *Nlrp3* (A350V*neoR*) in the myeloid lineage have spontaneous production of IL-1β and IL-18 and die in the neonatal period (Brydges *et al*, 2009). We then treated NLRP3-mutant mice with oral TR and found that TR-treated NLRP3 mutant mice had increased body weight at day 9 compared with the control group (Fig 4E and F). Moreover, oral TR treatment prevented the death of NLRP3 mutant mice (Fig 4G), suggesting that TR can suppress the lethal inflammation caused by NLRP3 mutation. Taken together, these results suggest that TR is active *in vivo* and can prevent NLRP3-dependent acute inflammation.

### TR has remarkable preventive or therapeutic effects on HFD-induced metabolic disorders

NLRP3 inflammasome has also been involved in chronic inflammation-associated complex diseases, including T2D (Martinon *et al*, 2006; Duewell *et al*, 2010; Masters *et al*, 2010; Zhou *et al*, 2010; Wen *et al*, 2011; Heneka *et al*, 2012; Lamkanfi & Dixit, 2012; Broderick *et al*, 2015), suggesting the possibility to reverse these diseases by inhibition of NLRP3 inflammasome-dependent inflammation. First, we tested whether TR was efficacious in preventing the development of metabolic disorders in a high-fat diet (HFD)-induced diabetic mouse model. The C57BL/6J mice were fed with HFD along with oral administration of TR at the dose of 25 mg/kg or 50 mg/kg. TR

treatment had a remarkable effect on the prevention of HFD-induced weight gain in a dose-dependent manner, although the food intake was no affected (Fig 5A and B). TR treatment also prevented the elevation of fasting glucose concentrations in HFD-treated mice (Fig 5C). In addition, the mice fed with TR showed better insulin sensitivity than the controls, as determined by glucose tolerance assays and insulin sensitivity assays (Fig 5D and E). We also evaluated whether TR had effects on HFD-induced hepatic steatosis and found that TR had striking effects on the appearance and size of liver organ (Appendix Fig S7A and B). Histological analyses of the mouse livers showed that although HFD induced massive hepatic steatosis, mice treated with TR showed little intracellular lipid accumulation in the liver (Appendix Fig S7C). However, TR treatment had no impact on weight gain, food intake, and blood glucose in normal diet-fed (chow) mice (Appendix Fig S8A–D), suggesting that TR is only functional in disease condition. Thus, these results indicate that TR has a remarkable preventive effect on HFD-induced metabolic disorders.

In diabetic mice, NLRP3 inflammasome-mediated low-grade metainflammation in metabolic organs, such as liver or adipose tissue, is critical for the development of T2D (Stienstra *et al*, 2010; Yan *et al*, 2013). To further confirm that TR prevents or treats metabolic disorders through inhibition of NLRP3 inflammasome activation, we tested whether TR treatment inhibited the NLRP3 inflammasome activation and metainflammation in diabetic mouse models. As expected, the serum, liver, or adipose tissues of HFD-treated mice showed increased IL-1β production, which were blocked by TR treatment (Fig 5F–G). The caspase-1 cleavage observed in HFD-treated mice was also suppressed by TR (Fig 5H), indicating that TR can inhibit metabolic stress-induced inflammasome activation *in vivo*. In addition, the production of TNF-α, which is an inflammasome-independent cytokine, was also suppressed in HFD-treated mice (Fig 5I), which was consistent with the observation in *Nlrp3*$^{-/-}$ mice (Yan *et al*, 2013). Thus, these results indicate that TR treatment can inhibit NLRP3 inflammasome activation and metainflammation in diabetic mice.

We further tested whether oral TR was effective in reversing metabolic disorders in the mouse model after the condition had been established. We first fed the mice with HFD for 14 weeks and then treated with oral TR for 6 weeks. TR treatment reversed the weight gain and significantly reduced the fasting or basal blood glucose concentrations in HFD-treated wild-type mice (Fig 6A–C). In addition, the TR-fed mice showed better insulin sensitivity than controls (Fig 6D and E). However, TR treatment had minimal effects on weight gain, blood glucose, and insulin sensitivity in HFD-fed *Nlrp3*$^{-/-}$ mice (Fig 6A–C, F and G), suggesting that TR improves diabetic symptoms *via* inhibition of NLRP3 inflammasome. Taken together, our results demonstrate that TR can reverse T2D-associated metabolic disorders by inhibition of NLRP3 inflammasome.

### TR is active *ex vivo* for cells from gouty patients

Since TR has remarkable beneficial effects on the mouse models of NLRP3-driven diseases, we then tested whether TR had effects on the pre-activated NLRP3 inflammasome on cells from patients with abnormal NLRP3 activation. Gout is an inflammatory arthritis caused by precipitation of monosodium urate (MSU) in articular

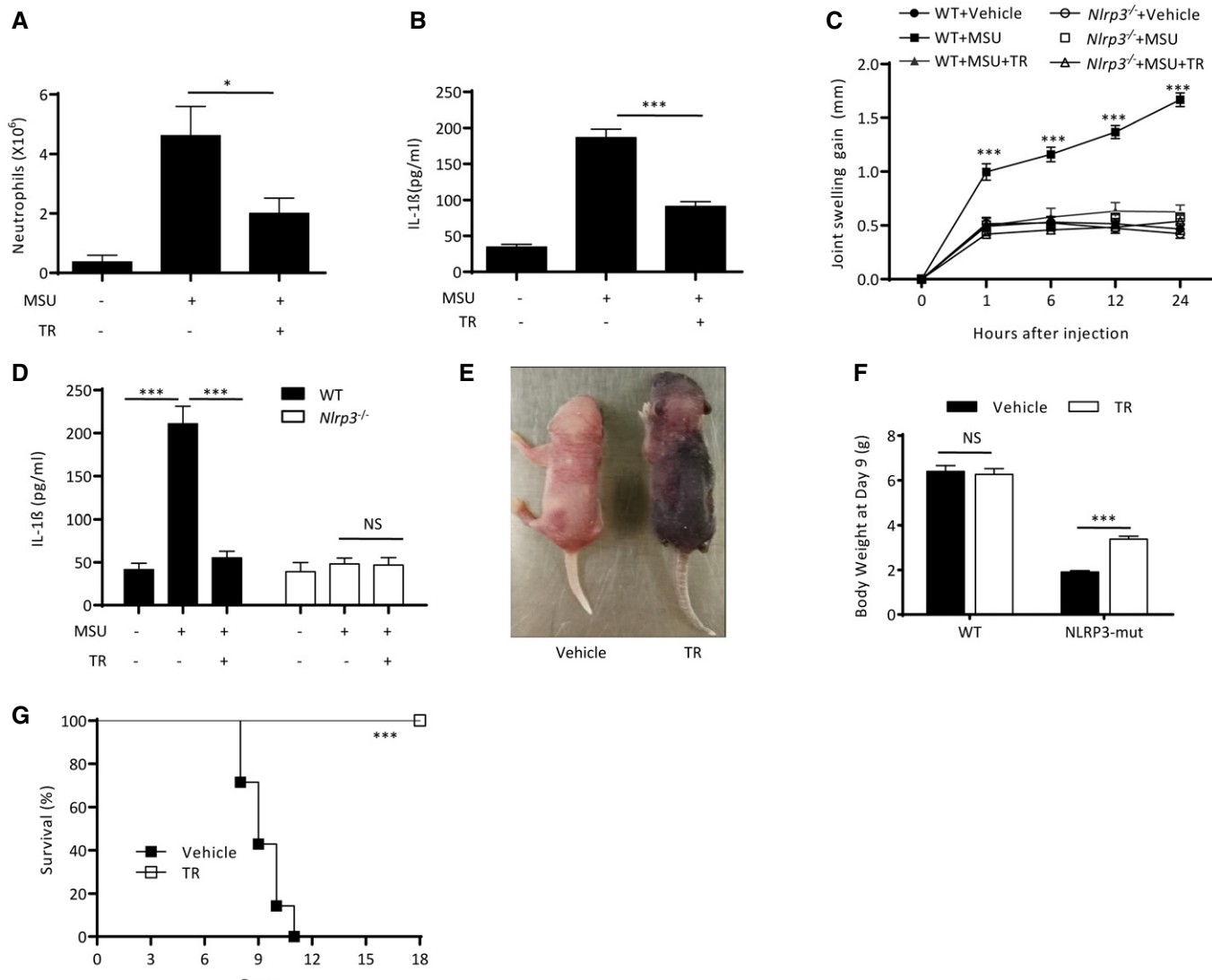

**Figure 4. TR inhibits NLRP3 activation *in vivo* and has preventive effects in mouse models of gouty arthritis and CAPS.**

A, B    FACS analysis of neutrophil numbers (A) or ELISA (B) of IL-1β in the peritoneal cavity of 10-week-old male C57BL/6J mice intraperitoneally injected with MSU (1 mg/mouse) with or without TR (200 mg/kg of body weight). *n* = 8 per group.

C, D    Time course of changes of joint swelling (C) or ELISA (D) of IL-1β in the supernatants of joint culture of 10-week-old male WT or *Nlrp3*⁻/⁻ mice intra-articularly injected with MSU with or without oral TR administration (200 mg/kg of body weight). *n* = 6 per group.

E    *Nlrp3A^{350VneoR}* crossed with LysM-Cre mice (NLRP3-mut) treated with vehicle or TR at day 9.

F    Weight of wild-type mice or *Nlrp3A^{350VneoR}* crossed with LysM-Cre mice (NLRP3-mut) treated with vehicle or TR at day 9. For WT vehicle and WT TR, *n* = 6; NLRP3-mut PBS *n* = 4; and NLRP3-mut TR *n* = 8.

G    Survival of NLRP3-mut mice treated with vehicle or TR up to day 18. TR group *n* = 8, vehicle group *n* = 7.

Data information: Data are shown as mean and s.e.m. and are representative of two or three independent experiments. Statistics were analyzed using an unpaired Student's *t*-test or a generalized Wilcoxon test (G): \*P < 0.05, \*\*\*P < 0.001, NS, not significant.

joints and bursal tissues of individuals with hyperuricemia (McQueen *et al*, 2012). NLRP3 inflammasome plays a critical role in MSU-induced inflammation, and clinical studies have demonstrated efficacy of IL-1 inhibitors in the treatment of patients with acute and chronic gout (Martinon *et al*, 2006; McGonagle *et al*, 2007; So *et al*, 2007; Terkeltaub *et al*, 2009; Jesus & Goldbach-Mansky, 2014). As expected, when the freshly isolated synovial fluid cells (SFCs) from a gouty patient were incubated without the stimulation of NLRP3 agonists, IL-1β secretion and caspase-1 processing could be detected

in the culture supernatants (Fig 7A and B). However, when these cells were incubated with the presence of TR, the caspase-1 activation and IL-1β production were inhibited in a dose-dependent manner (Fig 7A and B). In contrast, the production of inflammasome-independent cytokine TNF-α was not affected by TR treatment (Fig 7C). Thus, these results indicate that TR can suppress the pre-activated NLRP3 inflammasome in cells from patients and suggest that TR or its analogues might be used to control NLRP3-driven diseases in clinics.

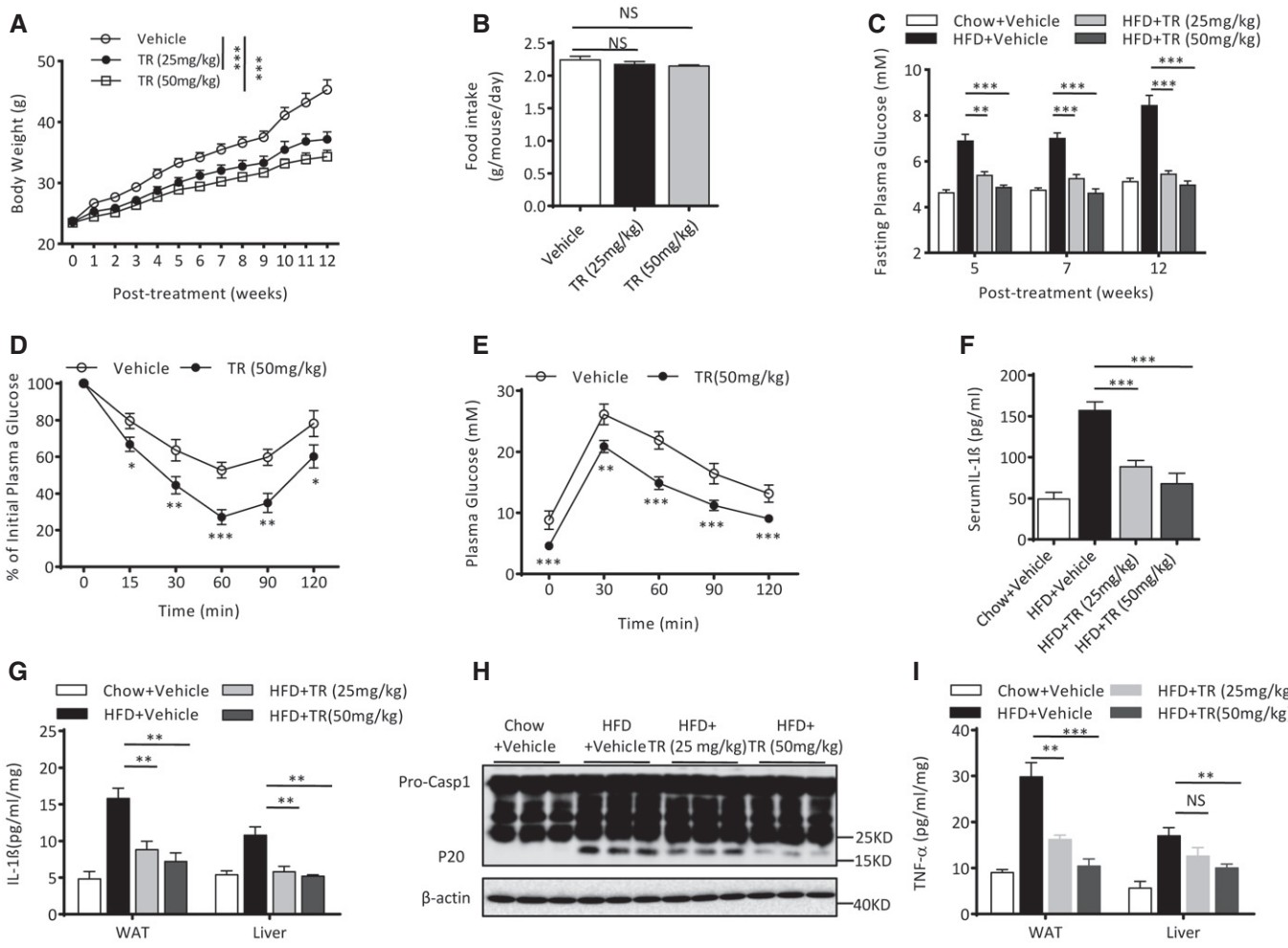

**Figure 5. The preventive role of TR in HFD-treated mice.**

A    Body weights of 6-week-old male wild-type mice were measured at the indicated time points after initiation of the HFD with or without oral TR treatment. *n* = 9 per group.

B    Daily food intake of the mice fed with high-fat diet (HFD) with or without oral TR treatment. *n* = 8 per group.

C    Fasting blood glucose concentrations at the indicated time points in mice fed with chow or HFD with indicated dose of oral TR. *n* = 7 per group.

D, E    Insulin tolerance test (ITT) (D) or glucose tolerance test (GTT) (E) performed at week 12 after initiation of the HFD with or without oral TR treatment. *n* = 7 per group.

F–I    Six-week-old male C57BL/6J mice were fed with chow or HFD for 12 weeks with or without treatment with different dose of TR. Serum IL-1β (F) was assessed by ELISA. Adipose tissue (WAT) and liver were isolated and cultured for 24 h, and supernatants were analyzed by ELISA for IL-1β (G) or TNF-α (I). Caspase-1 activation in WAT was analyzed by immunoblot as indicated (H). *n* = 7 per group.

Data information: Data are shown as mean and s.e.m. and are representative of two independent experiments. Statistics were analyzed using an unpaired Student's *t*-test or two-way ANOVA for (D and E): *$P < 0.05$, **$P < 0.01$, ***$P < 0.001$, NS, not significant.

Source data are available online for this figure.

# Discussion

Here, we demonstrate that TR, which is an old drug with high safety in clinic, has selective and potent inhibitory activity for NLRP3 inflammasome activation in mice *in vivo* and human cells *ex vivo*. TR might be a versatile small-molecule tool to study NLRP3 biology and its role in inflammatory diseases.

Our results demonstrate that TR directly targets NLRP3 to suppress inflammasome assembly. TR inhibited NLRP3 inflammasome activation, but had no effects on AIM2 or NLRC4 inflammasome activation, suggesting that it acts on the upstream of ASC to suppress inflammasome activation. TR treatment had no effects on potassium efflux, mitochondrial damage, and chloride efflux, which are critical steps or upstream events for NLRP3 activation (Martinon *et al*, 2009; Davis *et al*, 2011; Daniels *et al*, 2016), suggesting that it might directly target the NLRP3 inflammasome complex. Consistent with this, TR suppressed cLPS-induced IL-1β secretion, but could not inhibit the Gsdmd activation and pyroptosis. Indeed, the NLRP3 agonist-induced endogenous NLRP3-ASC, but not NLRP3–NEK7 interaction was inhibited by TR, suggesting that TR targets NLRP3 or NLRP3–ASC interaction to block the assembly of NLRP3 inflammasome. Moreover, TR could bind to the NACHT domain of

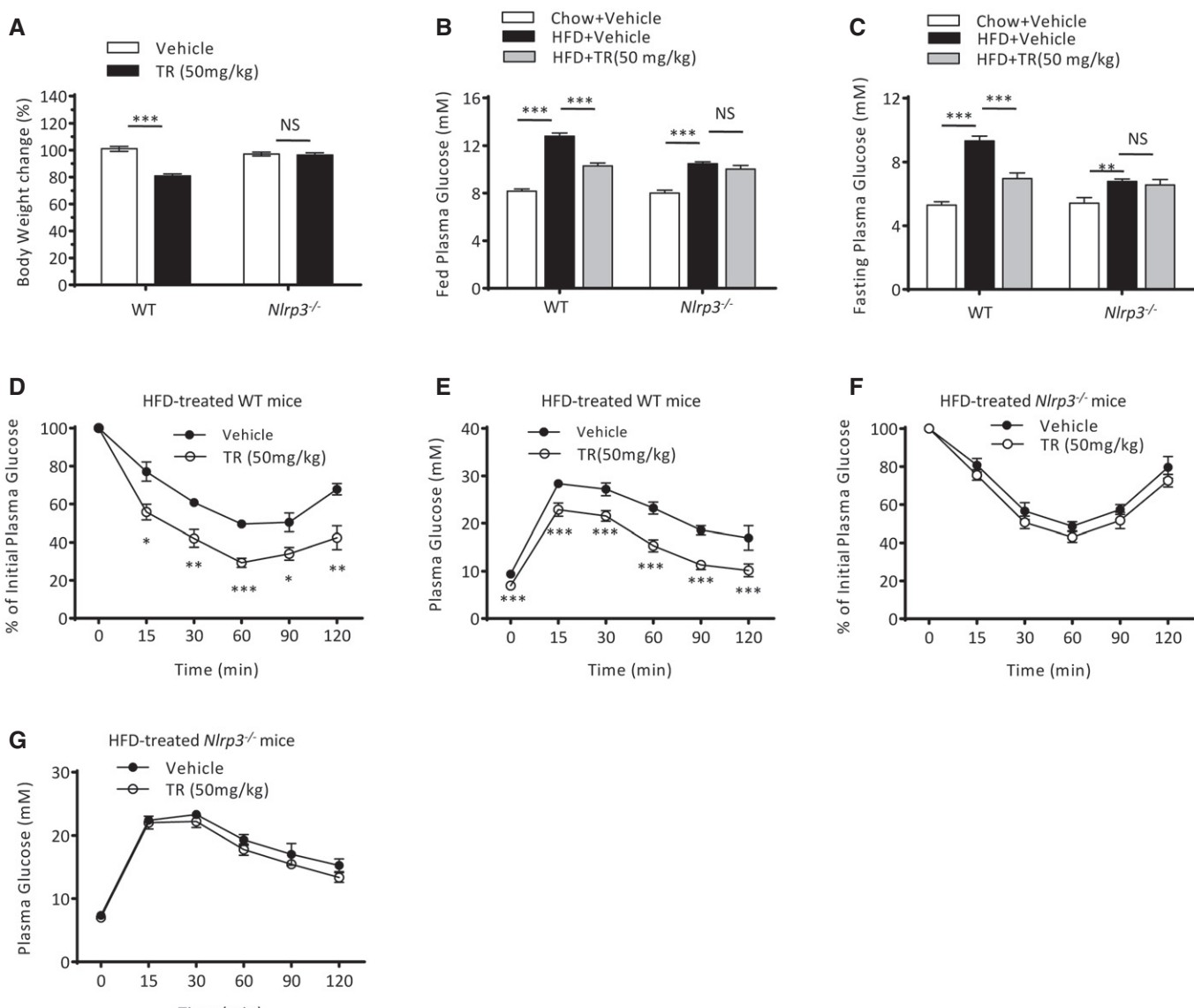

**Figure 6. Therapeutic role of TR in HFD-established diabetic mice.**

A    Body weights change of 6-week-old male wild-type (WT) or Nlrp3[−/−] mice that were first fed with HFD for 14 weeks and then treated with different dose of oral TR for 6 weeks. n = 6 per group.

B, C    Fed (B) or fasting (C) blood glucose concentrations at week 6 in the mice described in (A). n = 6.

D–G    ITT (D, F) or GTT (E, G) performed at week 6 in the mice described in (A). n = 6 per group.

Data information: Data are shown as mean and s.e.m. and are representative of two independent experiments. Statistics were analyzed using an unpaired Student's t-test or two-way ANOVA for (D–G): *P < 0.05, **P < 0.01, ***P < 0.001, NS, not significant.

NLRP3 and inhibit the direct NLRP3–NLRP3 interaction, suggesting that TR binds to NLRP3 and blocks NLRP3 oligomerization. Thus, our results provide the evidence showing that NLRP3 oligomerization could be targeted by small molecules to prevent or treat inflammasome-driven diseases.

An interesting question is that how a small molecule such as TR can inhibit NLRP3 oligomerization. Previous results have shown that the ATPase activity of NLRP3 NACHT domain is essential for the oligomerization of NLRP3 (Duncan et al, 2007). Moreover, several inhibitors, including parthenolide, Bay 11-7082, INF39,

3,4-methylenedioxy-β-nitrostyrene, and CY-09, have been reported to inhibit NLRP3 inflammasome activation by suppressing the ATPase activity of NLRP3 (Juliana et al, 2010; He et al, 2014; Cocco et al, 2017; Jiang et al, 2017), so it is possible that TR might inhibit the ATPase activity of NLRP3 to block its oligomerization. However, our data showed that TR had no effects on its ATPase activity. Another possibility is that TR might target the interfaces of NLRP3–NLRP3 interaction. Although the protein–protein interaction (PPI) interfaces are generally flat and large (roughly 1,000–2,000 A$^2$ per side) and are different with the deep cavities that typically bind

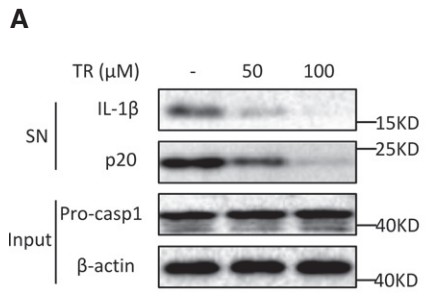
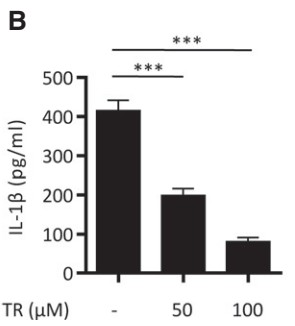
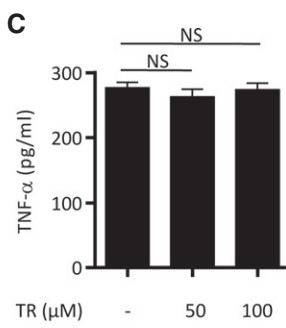

**Figure 7.   TR is active for human cells from gouty patients.**

A–C   Immunoblot analysis (A) of IL-1β and cleaved caspase-1 (p20) or ELISA of IL-1β (B), TNF-α (C) in supernatants from synovial fluid cells (SFCs) isolated from an individual with gout, treated with various doses of TR for 20 h. The SFCs isolated from four patients have been analyzed.

Data information: Data are shown as mean and s.e.m. and are representative of four independent experiments. Statistics were analyzed using an unpaired Student's t-test: ***$P < 0.001$, NS, not significant.

Source data are available online for this figure.

small molecules ($\sim$300–500 A$^2$) (Fuller et al, 2009; Hwang et al, 2010), not all residues at the PPI interface are critical (Arkin et al, 2014). Indeed, at least some PPIs might have small-molecule-sized "hot spots" that are essential for the interaction and can dynamically adjust to bind a small molecule (Arkin et al, 2014). In the last decade, more than 40 PPIs have now been targeted and several inhibitors have reached clinical trials (Labbe et al, 2013). So, TR might bind to a "hot spot" of NLRP3 that is critical for NLRP3–NLRP3 interaction and then block its activation. Future studies need to identify the residues of NLRP3 NACHT domain that are responsible for TR binding and clarify the detailed mechanism of how TR blocks NLRP3–NLRP3 interaction by using biochemical and structural approaches.

Although TR has shown a variety of biological activity, including anti-allergic and anti-inflammatory activity, the molecular targets of TR are not yet precisely known (Darakhshan & Pour, 2015). Our results demonstrate that NLRP3 inflammasome is the primary target for the action of TR in the mouse models of inflammation-associated diseases, at least in type 2 diabetes and gouty arthritis. Oral treatment of TR improved the symptoms of metabolic disorders or gouty arthritis in wild-type mice, but not in $Nlrp3^{-/-}$ mice, suggesting the beneficial effects of TR depend on its suppressive activity for NLRP3 inflammasome. However, it should be noted that TR had been reported to impair NF-κB activation and several other inflammasome-independent cytokines production, including IL-6, IL-8, and MCP-1 (Capper et al, 2000; Chikaraishi et al, 2001; Spiecker et al, 2002). Indeed, our results also showed that TR inhibited LPS-induced pro-IL-1β expression and IL-6 production in macrophages. It is possible that TR-induced inhibition of IL-1β downstream inflammatory signaling also contributes to the remarkable beneficial effects of TR.

T2D is a metabolic disorder that is characterized by high blood sugar and insulin resistance and can cause a number of complications, including heart disease, strokes, and diabetic retinopathy. However, the current available drugs are not effective in correcting the underlying cause of insulin resistance and most patients need pharmacotherapy in the rest of their lives (Nathan et al, 2009; Qaseem et al, 2012). Our study demonstrates that inhibition of

NLRP3-dependent metainflammation by TR is efficient to reverse the metabolic disorders in diabetic mice. Treatment with oral TR at the dose of 25 or 50 mg/kg/day (the equivalent human dose is 2.1 or 4.1 mg/kg/day) for about 6 weeks had dramatically therapeutic effects for both the hyperglycemia and insulin resistance in diabetic mice. In addition, we also found that treatment with TR could suppress hepatic steatosis in diabetic mice, suggesting that inhibition of NLRP3 inflammasome with TR can improve the dysfunction of both glucose and lipid metabolism. Thus, this study suggests that correcting NLRP3-dependent metainflammation might be an effective approach to treat T2D.

TR has been licensed for use in allergic disorders such as bronchial asthma in Japan and South Korea since 1982. Considering the high safety of TR in clinic, our study provides a potentially new and practical pharmacological approach for treating NLRP3-driven diseases, including CAPS, type 2 diabetes, and gout.

# Materials and Methods

### Mice

C57BL/6J mice used in the studies were obtained from Model Animal Research Center of Nanjing University. $Nlrp3^{-/-}$ mice were described previously (Martinon et al, 2006). LysM-cre mice (B6.129P2-$Lyz2^{tm1(cre)Ifo}$/J) and $Nlrp3^{A350VneoR}$ mice were from Jackson Laboratory. All animals were housed under 12-hr light/dark cycle at 22–24°C with unrestricted access to food and water for the duration of the experiment except during fasting tests. All animal experiments were approved by The Ethics Committee of University of Science and Technology of China.

### Reagents

MSU, Nigericin, ATP, PMA (phorbol-12-myristate-13-acetate), poly A:T, insulin, and glucose were purchased from Sigma. TR and MCC950 were obtained from Selleck. The ultrapure LPS and Pam3CSK4 (tripalmitoylcysteinylseryltetralysinelipopeptide) were

from Invivogen. Imject-Alum was from Pierce Biochemicals. Mito-Tracker and MitoSOX were from Invitrogen. Protein G agarose was from Millipore. Anti-β-actin (1:5,000, P30002) and Anti-DYKDDDDK-Tag mAb were from Abmart. Anti-human pro-IL-1β (1:1,000, 60136-1-Ig), anti-TRPV2 (1:1,000, 15991-1-AP), and anti-HPGDS (1:1,000, 22522-1-AP) were from Proteintech. Anti-mouse IL-1β (1:1,000, AF-401-NA) was from R&D Systems. Anti-mouse caspase-1 (p20) (1:1,000, AG-20B-0042) and anti-NLRP3 (1:1,000, AG-20B-0014) were from Adipogen. Anti-human caspase-1(1:1,000, 2225) was from Cell Signaling. Anti-ASC (1:500, sc-22514-R) and anti-NEK7 (1:500, SC-50756) were from Santa Cruz. Anti-human cleaved IL-1β (1:1,000, A5208206) was from Sangon Biotech. Anti-Flag (1:2,000, F2555) or anti-VSV (1:2,000, V4888) was from Sigma. Recombinant human NLRP3 was from Novus Biologicals. *Salmonella* is a gift from R.V. Bruggen.

## Human samples

The synovial fluid was obtained from four patients with gout and knee effusions. To use these clinical materials for research purposes, prior patients' written informed consents and approval from the Institutional Research Ethics Committee of Anhui Provincial Hospital were obtained (Approval No. 20160167).

Human data experiments conformed to the principles set out in the WMA Declaration of Helsinki and the Department of Health and Human Services Belmont Report.

## Cell preparation and stimulation

Bone marrow macrophages were derived from C57BL/6 mice and cultured in DMEM complemented with 10% FBS, 1 mM sodium pyruvate, and 2 mM L-glutamine in the presence of 20% culture supernatants of L929 mouse fibroblasts (CRL-6364). Human SFCs were isolated using Ficoll-Paque. THP-1, HEK-293T, and L929 cells were from American Type Culture Collection. THP-1, HEK-293T, L929 cells, and iBMDMs were not authenticated but routinely tested for mycoplasma contamination. THP-1 cells were differentiated for 4 h with 100 nM phorbol-12-myristate-13-acetate (PMA).

For induction of inflammasome activation, $5 \times 10^5$ macrophages were plated overnight in 12-well plates and the medium was changed to Opti-MEM (1% FBS) in the following morning, and then, the cells were primed for 3 h with ultrapure LPS (50 ng/ml) or Pam3CSK4 (400 ng/ml). After that, TR was added into the culture for another 30 min, and then, the cells were stimulated for 4 h with MSU (150 μg/ml), Alum (300 μg/ml), *S. typhimurium* (multiplicity of infection, MOI) or for 30 min with ATP (2.5 mM) or nigericin (3 μM). Cells were transfected with poly A:T (0.5 μg/ml) for 4 h or LPS (500 ng/ml) for overnight through the use of Lipofectamine 2000 according to the manufacturer's protocol (Invitrogen). Cell extracts and precipitated supernatants were analyzed by immunoblot.

## Immunofluorescence

BMDMs were plated on coverslips with the density of $2–3 \times 10^5$/ml for overnight. The following day, the medium was replaced with Opti-MEM (1% FBS) containing LPS (50 ng/ml) for 3 h, and then, the indicated inhibitors or controls were added for 30 min. After that, BMDMs were used for stimulation and staining with

MitoTracker Red (50 nM) or Mitosox (5 μM). After washing the cells three times with PBS, the cells were fixed with 4% PFA in PBS for 15 min at room temperature and then were washed three times with PBST. Confocal microscopy analyses were carried out using a Zeiss LSM700.

## Generation of *Gsdmd*$^{-/-}$ iBMDMs and Flag-Gsdmd reconstitution

To construct *Gsdmd*$^{-/-}$ iBMDMs, gRNAs (AGCATCCTGGCATTCCG AG) were transduced into Cas9$^+$ cloned iBMDM lines by lentiviral delivery with lentiCRISPRv2 (Addgene) followed by selection of gRNA-expressing cells by Puromycin (Life Technologies). To reconstitute Flag-Gsdmd, the *Gsdmd*$^{-/-}$ iBMDMs were lentivirally transduced with cDNA encoding mouse *Gsdmd* using pLEX vector (Thermo Fisher).

## SiRNA-mediated gene silences in BMDMs

BMDMs were plated in 12-well plates (at a density of $3 \times 10^5$ cells per well) and then were transfected with 50 nM siRNA through the use of Lipofectamine RNAiMAX according to the manufacturer's guidelines (Invitrogen). SiRNA sequences were chemically synthesized by GenePharma, and the negative control siRNA was also from GenePharma. The siRNA sequences were as follows: siTRPV2 (5-GCTGGCTGAACCTGCTTTATT-3); siHPGDS (5-CCUAACUACAAA CUGCUUUTT-3).

## ELISA

Supernatants from cell culture, tissue culture, or serum were assayed for mouse IL-1β, IL-6, TNF-α, and human IL-1β or TNF-α (R&D Systems) according to the manufacturer's instructions.

## MSU-induced peritonitis and arthritis

Ten-week-old male C57BL/6 mice were injected intraperitoneally (i.p.) with different doses of TR or MCC950 30 min before i.p. injection of MSU (1 mg MSU crystals dissolved in 0.5 ml sterile PBS). After 6 h, mice were killed and peritoneal cavities underwent lavage with 10 ml cold PBS. Peritoneal lavage fluid was assessed by flow cytometry (BD) with the neutrophil markers Ly6G and CD11b for analysis of the recruitment of polymorphonuclear neutrophils and determined IL-1β production by ELISA.

For inducing joint inflammation, 10-week-old male C57BL/6J mice were orally administered TR (200 mg/kg, dissolved in sodium carboxymethylcellulose, 0.5%). After 1 h, MSU (0.5 mg MSU crystals dissolved in 20 μl sterile PBS) was administrated by intra-articular injection. The size of joints was measured with an electronic caliper at the indicated time points. Twenty-four h after intra-articular injection, the patella were isolated from inflamed knee joints and cultured for 1 h at room temperature in Opti-MEM containing 1% penicillin–streptomycin (200 μl/patella). Protein levels of murine IL-1β were measured by ELISA.

## Muckle–Wells syndrome mouse model

*Nlrp3*$^{A350VneoR}$ mice were crossed with LysM-Cre mice (B6.129P2-*Lyz2*$^{tm1(cre)Ifo}$/J). TR was administered orally (100 mg/kg) twice a

day starting at day 4 after birth. The weight and survival of mice were monitored every day.

### Determination of intracellular potassium or chloride

For accurate measurement of the intracellular potassium, BMDMs were plated overnight in 6-well plates and then primed with 50 ng/ml LPS for 3 h. After that, cells were treated with TR for 30 min and then stimulated with nigericin for 30 min. Culture medium was thoroughly aspirated and lysed with 3% ultrapure HNO3. Intracellular $K^+$ measurements were performed by inductively coupled plasma optical emission spectrometry with a PerkinElmer Optima 2000 DV spectrometer using yttrium as the internal standard.

For accurate measurement of the intracellular chloride, BMDMs were plated overnight in 12-well plates and then primed with 50 ng/ml LPS for 3 h. After that, cells were treated with TR for 30 min and then stimulated with nigericin for 15 min. The supernatants of 12-well plates were removed, and ddH$_2$O was added (200 μl per well) and kept for 15 min at 37°C. The lysates were transferred to 1.5-ml EP tube and centrifuged at $10,000 \times g$ for 5 min. 50 μl supernatants was then transferred the to a new 1.5-ml EP tube and mixed with 50 μl MQAE (10 μM). The absorbance was tested by BioTek Multi-Mode Microplate Readers (Synergy2). A control was settled in every experiment to determine the extracellular amount of chloride remaining after aspiration, and this value was subtracted.

### Immunoprecipitation and pull-down assay

For the endogenous interaction assay, BMDMs were stimulated and lysed with NP-40 lysis buffer with complete protease inhibitor. The cell lysates were incubated overnight at 4°C with the primary antibodies and Protein G Mag Sepharose (GE Healthcare). The proteins bound by antibody were pulled down by protein G beads and subjected to immunoblotting analysis. For the exogenous interaction assay, HEK-293T cells ($3 \times 10^5$/ml) were transfected with plasmids in 6-well plates via the polyethylenimine. After 24 h, cells were collected and lysed with NP-40 lysis buffer with complete protease inhibitor. Extracts were immunoprecipitated with anti-Flag antibody and beads and then were assessed by immunoblot analysis.

For pull-down assay, BMDMs or 293T lysates were collected and centrifuged at $6,200 \times g$. The supernatant was transferred to another tube and the cell debris was thoroughly discarded. Prewashed streptavidin beads were added into the supernatant, allowing 2-h pre-incubation with motion at 4°C to remove unspecific binding proteins. Purified human recombinant NLRP3 proteins were dissolved in lysis buffer. The lysates or recombinant proteins were incubated with compounds, followed by incubation with indicated doses of biotin-TR for 1 h. After that, the samples were then incubated with streptavidin beads for 2 h. Beads were washed with 0.1% Tween-20 in PBS and 1% NP-40 in PBS and boiled in SDS buffer.

### ASC oligomerization assay

BMDMs were seeded at $1 \times 10^6$/ml in 6-well plates. The following day, the medium was replaced and cells were primed with 50 ng/ml LPS for 3 h. The cells were treated with TR for 30 min and then stimulated with nigericin for 30 min. The supernatants were removed, cells were rinsed in ice-cold PBS, and then, cells were lysed by NP-40 for 30 min. Lysates were centrifuged at $330 \times g$ for 10 min at 4°C. The pellets were washed twice in 1 ml of ice-cold PBS and resuspended in 500 μl of PBS. 2 mM disuccinimydylsuberate (DSS) was added to the resuspended pellets, which were incubated at room temperature for 30 min with rotation. Samples were then centrifuged at $330 \times g$ for 10 min at 4°C. The cross-linked pellets were resuspended in 30 μl sample buffer and then were boiled and analyzed by immunoblot.

### Semi-denaturing detergent agarose gel electrophoresis (SDD-AGE)

The oligomerization of NLRP3 was analyzed according to the published protocol (Hou et al, 2011). Cells were lysed with Triton X-100 lysis buffer (0.5% Triton X-100, 50 mM Tris–Hcl, 150 mM NaCl, 10% glycerol, 1 mM PMSF, and protease inhibitor cocktail), which were then resuspended in 1× sample buffer (0.5× TBE, 10% glycerol, 2% SDS, and 0.0025% bromophenol blue) and loaded onto a vertical 1.5% agarose gel. After electrophoresis in the running buffer (1× TBE and 0.1% SDS) for 1 h with a constant voltage of 80 V at 4°C, the proteins were transferred to Immobilon membrane (Millipore) for immunoblotting. 1× TBE buffer contains 89 mM Tris (pH 8.3), 89 mM boric acid, 2 mM EDTA.

### NLRP3 ATPase activity assay

Purified recombinant human NLRP3 (Novus Biologicals) was incubated at 37°C with different concentrations (25, 50, or 100 μM) of TR for 15 min in the reaction buffer. ATP (250 μm, Ultra Pure ATP) was then added, and the mixture was further incubated at 37°C for another 40 min. The amount of ATP converted into adenosine diphosphate (ADP) was determined by luminescent ADP detection with ADP-Glo Kinase Assay Kit (Promega, Madison, MI, USA) according to the manufacturer's protocol. The results were expressed as percentage of residual enzyme activity to the vehicle-treated enzyme.

### Lactate dehydrogenase assay

The release of lactate dehydrogenase (LDH) into the culture medium was determined by LDH Cytotoxicity Assay Kit (Thermo Fisher) according to the manufacturer's instructions.

### High-fat diet treatments

At the age of 6 weeks, male C57BL/6J mice with similar plasma glucose levels and body weights were randomized into different groups. For prevention research, C57BL/6J mice were fed with 60 kcal% fat diet (HFD, Medicience Ltd, China) until the end of the experiments. For generation of HFD-induced diabetic mice, $Nlrp3^{-/-}$ mice and littermate controls (WT) were fed with HFD for 14 weeks and then maintained with HFD when used for the subsequent experiments. TR (25 or 50 mg/kg of body weight) was brought into suspension in sodium carboxymethylcellulose (Na-CMC, 0.5%) and orally administered 2 h before dark cycle once a day during experiments.

**The paper explained**

**Problem**
NLRP3 inflammasome contributes to the development of a wide variety of human diseases, including gout, atherosclerosis, neurodegenerative diseases and type 2 diabetes (T2D), but the *medications* targeting NLRP3 inflammasome are not available in clinic.

**Results**
Here, we show that tranilast (TR), an old anti-allergic clinical drug, can specifically inhibit NLRP3 inflammasome activation in both human and mouse macrophages. TR directly binds to NLRP3 and suppresses its oligomerization and the subsequent NLRP3 complex formation. *In vivo*, TR has remarkable preventive or therapeutic effects on the mouse models of NLRP3 inflammasome-related human diseases, including gouty arthritis, cryopyrin-associated autoinflammatory syndromes, and type 2 diabetes.

**Impact**
Our study identifies NLRP3 as a direct target of the old drug TR and provides a potentially practical pharmacological approach for treating NLRP3-driven diseases.

**Blood glucose assay**

Glucose levels in blood collected from the tail vein were determined using a One Touch® Ultra® Blood Glucose Test System Kit (Lifespan Company, USA).

**Glucose tolerance or insulin tolerance test**

Glucose tolerance (GTT) was performed via intraperitoneal (i.p.) injection of glucose at 1.5 g/kg for C57BL/6J mice after a 14-h fasting from the beginning of dark cycle. ITT was performed *via* i.p. injection of human recombinant insulin (Novo Nordisk) at dose of 1 IU/kg for C57BL/6J mice after 4-h fasting. Blood glucose levels were measured from the tail veil at 15, 30, 60, 90, and 120 min following glucose or insulin injection.

**Mouse liver histological analysis**

Mouse liver tissues were post-fixed in 4% PFA for 24 h under 4°C conditions and sectioned after embedding in paraffin. The sections were prepared and stained with H&E or Oil red O using standard procedures. Slides were examined under a Nikon ECLIPSE Ci biological microscope, and images were captured with a NikonDS-U3 color digital camera.

**Statistical analyses**

All values are expressed as the mean and s.e.m. Statistical analysis was performed with the unpaired *t*-test for two groups or two-way ANOVA (GraphPad Software or Excel software) using for multiple groups with all data points showing a normal distribution. No exclusion of data points was used. The researchers were not blinded to the distribution of treatment groups when performing experiments and data assessment. Sample sizes were selected on the basis of preliminary results to ensure an adequate power. *P*-values

$< 0.05$ were considered significant. All *P*-values are shown in Appendix Table S1.

**Expanded View** for this article is available online.

## Acknowledgements
We thank Dr. Jurg Tschopp (University of Lausanne) for providing $Nlrp3^{-/-}$ mice. This work was supported by the National Basic Research Program of China (2014CB910800), the National Natural Science Foundation of China (81788101, 91742202, 81525013, 81722022, 81701549, 31700777), the Strategic Priority Research Program of the Chinese Academy of Sciences (XDA12040310, XDPB0303), the Young Talent Support Program, and the Fundamental Research Funds for the Central Universities.

## Author contributions
YH, HJ, YC, XW, and YY performed the experiments of this work; HJ, JT, XD, GL, HZ, WJ, and RZ designed the research. YH, WJ, and RZ wrote the manuscript. WJ and RZ supervised the project.

## Conflict of interest
The authors declare that they have no conflict of interest.

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
