## [Review Process File · EMBO Molecular Medicine]

Tranilast directly targets NLRP3 to treat inflammasome-driven diseases

Yi Huang, Hua Jiang, Yun Chen, Xiaqiong Wang, Yanqing Yang, Jinhui Tao, Xianming Deng, Gaolin Liang, Huafeng Zhang, Wei Jiang, Rongbin Zhou

Review timeline:

Submission date:	14 November 2017
Editorial Decision:	14 December 2017
Revision received:	23 January 2018
Editorial Decision:	6 February 2018
Revision received:	10 February 2018
Accepted:	19 February 2018

Editor: Céline Carret

Transaction Report:

1st Editorial Decision

14 December 2017

Thank you for the submission of your manuscript to EMBO Molecular Medicine. We have now heard back from the three referees whom we asked to evaluate your manuscript.

You will see from the set of comments pasted below that overall the referees are positive about the paper. This said, in order to increase clarity and conclusiveness, you are strongly encouraged to do the following: compare MCC950 to TR to show how it performs (ref.1), test TR effect on pyroptosis (ref.2), and provide details and clarifications throughout including full western blots, expand discussion and proofread the text (ref2 and 3). During our cross-commenting exercise, it became clear that all referees support the requested drugs comparison, and I really hope you'll be able to perform it.

We would welcome the submission of a revised version within three months for further consideration and would like to encourage you to address all the criticisms raised as suggested to improve conclusiveness and clarity. Please note that EMBO Molecular Medicine strongly supports a single round of revision and that, as acceptance or rejection of the manuscript will depend on another round of review, your responses should be as complete as possible.

I look forward to receiving your revised manuscript.

***** Reviewer's comments *****

Referee #1 (Comments on Novelty/Model System for Author):

In this work Huang et al describe Tranilast as a specific NLRP3 inhibitor, and provides mechanistic data as well as in vivo data demonstrating a strong effects in mouse models of known NLRP3-dependent diseases. The work is of very high quality, and the conclusions are supported by the data. However, as the data stand now, it is difficult to critically assess how potent Tranilast is compared to other known NLRP3 antagonists, e.g. MCC950. This should be tested, both in vitro and in vivo.

Referee #1 (Remarks for Author):

Very nice piece of work. My only point (which I think is important to be able to put the data in context with the rest of the field), is that more data - both in vitro and in vivo - should be provided where Tranilast is compared directly to other known NLRP3 antagonists.

Referee #2 (Remarks for Author):

In this manuscript, the authors report that Tranilast (TR), an old anti-allergic clinical drug, specifically inhibits NLRP3 inflammasome activation in macrophages by inhibiting the assembly of NLRP3 inflammasome. They show that TR directly binds to NLRP3 and prevents its oligomerization. With in vivo experiments, the author further reports that TR has remarkable preventive or therapeutic effects on mouse models of NLRP3 inflammasome-related human diseases as gouty arthritis, cryopyrin-associated autoinflammatory syndrome and type 2 diabetes. Finally they show that TR is also efficient *ex vivo* using mononuclear cells from patients with Gout. Hence, as a direct NLRP3 inhibitor, TR appears as a promising component for treating NLRP3-driven diseases.

The data are very convincing and the conclusions raised by the authors are well supported by their results. My only concern is that the authors have not investigated the impact of TR on pyroptosis. Indeed, NLRP3 inflammasome activation triggers caspase-1 processing and this inflammatory caspase not only promotes the maturation of IL-1 β and IL-18 but also cleaves gasdermin D to promote pyroptosis and ensuing the release of the pro-inflammatory cytokines. As TR prevents NLRP3 inflammasome activation, a decrease in pyroptosis/cell death is expected in TR-pretreated cells after exposure to nigericin, ATP, MSU and alum. This should be shown. Likewise, an inhibition in the gasdermin D cleavage is expected. In the case of cLPS, in contrast, the pre-treatment with TR should have a minor effect on gasdermin D cleavage and ensuing pyroptosis as in this condition, NLRP3 inflammasome activation is a consequence of the K⁺ potassium efflux triggered by the pyroptosis after the cleavage of gasdermin D by caspase-4 (in human) or caspase-1 (in mouse).

Other comments:

In Figure 5H, the authors show a full WB of caspase-1 in cell extract where pro-caspase-1 and the different isoforms as well as the p20 mature form can be seen. I do not understand why in the other figures, in the "input", the authors only show the pro-caspase-1 given that after strong signals as exposure to nigericin, the p20 caspase-1 can be seen in cell extract. Likewise, for pro-IL-1 β , after priming in BMDMs, several isoforms of IL-1 β are detected in cell extracts and after NLRP3 inflammasome activation, the mature p17 IL-1 β is very often detected.

Referee #3 (Comments on Novelty/Model System for Author):

This is a study of significant novelty and translational value. Yet, the study lacks essential experimental information throughout. Also, the English used needs polishing.

Referee #3 (Remarks for Author):

Huang et al. have investigated the effect of Tranilast (TR), an old anti-allergic clinical drug, in inflammasome activation. They found that TR directly inhibits NLRP3 activation by binding to the NACHT domain of NLRP3 and suppressing the assembly of the fully functional NLRP3 inflammasome. This seems to be specific for NLRP3 as the activation of AIM2 and NLRC4 inflammasomes is not affected. They also show that in experimental animal models *in vivo* TR suppresses gouty arthritis, cryopyrin-associated autoinflammatory syndromes and type 2 diabetes. Overall, this is an interesting and novel study. As inhibitors suppressing NLRP3 activation of clinical applicability are actively being sought, this study has also significant translational potential.

Specific comments

1. Often, information on essential experimental details is lacking in both the text and Fig. legend,

making difficult the assessment of the data. For example, in Fig. 1, there is no information provided about the duration of LPS stimulation in the various situations, the timing of supernatant or cell extract collection for ELISA etc. In Fig. 2 and 3 the timing of treatment with nigericin is not mentioned. This goes throughout the manuscript. This is essential information and should be part of the text and Figure legends.

2. In lines 127-128, the authors state that they searched for 'NLRP3 inhibitors in clinical drugs and found TR can directly...'. If they include that as an approach they used in this work, they should also have the data in.

3. The word 'priming' as used in lines 180-191 to describe the results of Fig. 1 is inappropriate and confusing. LPS induces 'priming' of the inflammasome anyway, and that is independent of the addition of TR before or after LPS stimulation. Thus, TR treatment at 3h post LPS stimulation should not be affecting priming (much) while treatment before should.

4. In Fig. 2, B-C it is not clear what IP on the left and IP on the top indicates. The NLRP3, ASC and b-actin WB at the bottom (labelled on the left as 'input') are not extracts before IP? Also, in Fig. 3 is the I.P. in this case called 'pulldown'? The same wording should be used.

5. How do the authors envisage a small molecule such as TR inhibiting NLRP3 oligomerization? Also, how can that be specific for NLRP3 but not other inflammasomes? Moreover, how can a small molecule such as TR inhibit protein interactions driven by the interaction of large protein surfaces?

6. The Discussion is short and should be expanded

7. English needs polishing.

1st Revision - authors' response

23 January 2018

Referee #1 (Comments on Novelty/Model System for Author):

In this work Huang et al describe Tranilast as a specific NLRP3 inhibitor, and provides mechanistic data as well as *in vivo* data demonstrating a strong effects in mouse models of known NLRP3-dependent diseases. The work is of very high quality, and the conclusions are supported by the data. However, as the data stand now, it is difficult to critically assess how potent Tranilast is compared to other known NLRP3 antagonists, e.g. MCC950. This should be tested, both *in vitro* and *in vivo*.

Referee #1 (Remarks for Author):

Very nice piece of work. My only point (which I think is important to be able to put the data in context with the rest of the field), is that more data - both in vitro and in vivo - should be provided where Tranilast is compared directly to other known NLRP3 antagonists.

Reply: Thanks very much for the suggestion. We have compared the activity of TR with MCC950 and the data were shown as appendix Fig.S6A-C in the revised manuscript. The results showed that although the *in vitro* inhibitory activity of TR on MSU-induced IL-1b secretion was around 100-500 times less potent than MCC950 (Appendix Fig S6A), its *in vivo* activity on MSU-induced peritonitis was only around 5-10 times less potent than MCC950 (Appendix Fig S6B, C).

Referee #2 (Remarks for Author):

In this manuscript, the authors report that Tranilast (TR), an old anti-allergic clinical drug, specifically inhibits NLRP3 inflammasome activation in macrophages by inhibiting the assembly of NLRP3 inflammasome. They show that TR directly binds to NLRP3 and prevents its oligomerization. With in vivo experiments, the author further report that TR has remarkable preventive or therapeutic effects on mouse models of NLRP3 inflammasome-related human diseases as gouty arthritis, cryopyrin-associated autoinflammatory syndrome and type 2 diabetes. Finally they show that TR is also efficient ex vivo using mononuclear cells from patients with Gout. Hence, as a direct NLRP3 inhibitor, TR appears as a promising component for treating NLRP3-driven diseases.

The data are very convincing and the conclusions raised by the authors are well supported by their results. My only concern is that the authors have not investigated the impact of TR on pyroptosis. Indeed, NLRP3 inflammasome activation triggers caspase-1 processing and this inflammatory caspase not only promotes the maturation of IL-1 β and IL-18 but also cleaves gasdermin D to promote pyroptosis and ensuing the release of the pro-inflammatory cytokines. As TR prevents NLRP3 inflammasome activation, a decrease in pyroptosis/cell death is expected in TR-pretreated cells after exposure to nigericin, ATP, MSU and alum. This should be shown. Likewise, an inhibition in the gasdermin D cleavage is expected. In the case of cLPS, in contrast, the pre-treatment with TR should have a minor effect on gasdermin D cleavage and ensuing pyroptosis as in this condition; NLRP3 inflammasome activation is a consequence of the K⁺ potassium efflux triggered by the pyroptosis after the cleavage of gasdermin D by caspase-4 (in human) or caspase-11 (in mouse).

Reply: Thanks very much for the suggestions. In the revised manuscripts, we provided new data showing that TR could block nigericin-induced pyroptosis and Gsdmd activation (Appendix Fig S1A, S1E), but could not block cLPS-induced pyroptosis and Gsdmd activation (Appendix Fig S1E, S1F).

Other comments:

In Figure 5H, the authors show a full WB of caspase-1 in cell extract where pro-caspase-1 and the different isoforms as well as the p20 mature form can be seen. I do not understand why in the other figures, in the "input", the authors only show the pro-caspase-1 given that after strong signals as exposure to nigericin, the p20 caspase-1 can be seen in cell extract. Likewise, for pro-IL-1 β , after priming in BMDMs, several isoform of IL-1 β are detected in cell extracts and after NLRP3 inflammasome activation, the mature p17 IL-1 β is very often detected.

Reply: Thanks very much for the suggestion. In our experiences, the caspase-1 P20 and IL-1 β p17 were weak in the cell extracts of BMDMs when stimulated with NLRP3 activators (please see the data below), so we only showed the pro-caspase-1 and pro-IL-1 β in the "input". Indeed, most publications in this field showed the data in a similar way.

Referee #3 (Comments on Novelty/Model System for Author):

This is a study of significant novelty and translational value. Yet, the study lacks essential experimental information throughout. Also, the English used needs polishing.

Referee #3 (Remarks for Author):

Huang et al. have investigated the effect of Tranilast (TR), an old anti-allergic clinical drug, in inflammasome activation. They found that TR directly inhibits NLRP3 activation by binding to the NACHT domain of NLRP3 and suppressing the assembly of the fully functional NLRP3 inflammasome. This seems to be specific for NLRP3 as the activation of AIM2 and NLRC4 inflammasomes is not affected. They also show that in experimental animal models in vivo TR suppresses gouty arthritis, cryopyrin-associated autoinflammatory syndromes and type 2 diabetes. Overall, this is an interesting and novel study. As inhibitors suppressing NLRP3 activation of clinical applicability are actively being sought, this study has also significant translational potential.

Specific comments

1. Often, information on essential experimental details is lacking in both the text and Fig. legend, making difficult the assessment of the data. For example, in Fig. 1, there is no information provided about the duration of LPS stimulation in the various situations, the timing of supernatant or cell extract collection for ELISA etc. In Fig. 2 and 3 the timing of treatment with nigericin is not mentioned. This goes throughout the manuscript. This is essential information and should be part of the text and Figure legends.

Reply: Thanks very much for the suggestions. Some details have been described in methods sections of the original manuscripts. For example, we have provided the information about inflammasomes stimulation: "For induction of inflammasome activation, 5×10^5 macrophages were plated overnight in 12-well plates and the medium was changed to Opti-MEM (1% FBS) in the following morning, then the cells were primed for 3 h with ultrapure LPS (50 ng/ml) or Pam3CSK4 (400 ng/ml). After that, TR were added into the culture for another 30 min and then the cells were stimulated for 4 h with MSU (150 μ g/ml), Alum (300 μ g/ml), *S. typhimurium* (multiplicity of infection (MOI)) or for 30 min with ATP (2.5 mM) or nigericin (3 μ M). Cells were transfected with poly A:T (0.5 μ g/ml) for 4 h or LPS (500 ng/ml) for overnight through the use of Lipofectamine2000 according to the manufacturer's protocol (Invitrogen)."

In the revised manuscripts, we added some lacking information in figure legends or Methods.

2. *In lines 127-128, the authors state that they searched for 'NLRP3 inhibitors in clinical drugs and found TR can directly...'. If they include that as an approach they used in this work, they should also have the data in.*

Reply: Thanks for the comments. We revised the sentence as "In this study, we showed that TR directly bound to NLRP3 and inhibited NLRP3 inflammasome assembly and the subsequent caspase-1 activation and IL-1 β production" in the revised manuscript.

3. *The word 'priming' as used in lines 180-191 to describe the results of Fig. 1 is inappropriate and confusing. LPS induces 'priming' of the inflammasome anyway, and that is independent of the addition of TR before or after LPS stimulation. Thus, TR treatment at 3h post LPS stimulation should not be affecting priming (much) while treatment before should.*

Reply: Thanks very much for the comment. We revised the sentences as "we then examined whether TR inhibited NLRP3 inflammasome activation via regulating the expression of NF- κ B-dependent NLRP3 or pro-IL-1 β expression. When BMDMs were stimulated with TR after 3-hour LPS treatment, TR had no effect on LPS-induced NLRP3, pro-IL-1 β expression, TNF- α or IL-6 production (Fig 1C, D and appendix Fig S1B-D), suggesting that TR-induced NLRP3 inflammasome inhibition was not caused by the downregulation of NLRP3 or pro-IL-1 β expression at this condition".

4. *In Fig. 2, B-C it is not clear what IP on the left and IP on the top indicates. The NLPR3, ASC and b-actin WB at the bottom (labelled on the left as 'input') are not extracts before IP? Also, in Fig.3 is the I.P. in this case called 'pulldown'? The same wording should be used.*

Reply: Thanks for the comments. The NLPR3, ASC and b-actin WB at the bottom (labelled on the left as 'input') are extracts before IP. We corrected the labels on the top as "IgG" and "anti-NEK7 or ASC" In Fig.2B, C.

In The Fig.3, streptavidin beads (not antibodies) were used to pulldown, so in my opinion, they are not classical "Immunoprecipitation" and "Pulldown" seems to be better than "IP" in these figures.

5. *How do the authors envisage a small molecule such as TR inhibiting NLRP3 oligomerization? Also, how can that be specific for NLRP3 but not other inflammasomes? Moreover, how can a small molecule such as TR inhibit protein interactions driven by the interaction of large protein surfaces?*

Reply: It is a good question. Honestly we don't know the detailed mechanism of how TR inhibits NLRP3 oligomerization. We have discussed this issue in the revised manuscript: "An interesting question is that how a small molecule such as TR can inhibit NLRP3 oligomerization. Previous results have shown that the ATPase activity of NLRP3 NACHT domain is essential for the oligomerization of NLRP3 (Duncan et al, 2007). Moreover, several inhibitors, including parthenolide, Bay 11-7082, INF39, 3,4-methylenedioxy- β -nitrostyrene and CY-09 have been reported to inhibit NLRP3 inflammasome activation by suppressing the ATPase activity of NLRP3 (Cocco et al, 2017; He et al, 2014; Jiang et al, 2017; Juliana et al, 2010), so it is possible that TR might inhibit the ATPase activity of NLRP3 to block its oligomerization. However, our data showed that TR had no effects on its ATPase activity. Another possibility is that TR might target the interfaces of NLRP3-NLRP3 interaction. Although the protein-protein interaction (PPI) interfaces are generally flat and large (roughly 1,000–2,000 \AA^2 per side) and are different with the deep

cavities that typically bind small molecules ($\sim 300\text{--}500 \text{ \AA}^2$) (Fuller et al, 2009; Hwang et al, 2010), not all residues at the PPI interface are critical (Arkin et al, 2014). Indeed, at least some PPIs might have small-molecule-sized "hot spots" that are essential for the interaction and can dynamically adjust to bind a small molecule (Arkin et al, 2014). In the last decade, more than 40 PPIs have now been targeted and several inhibitors have reached clinical trials (Labbe et al, 2013). So, TR might bind to a "hot spot" of NLRP3 that is critical for NLRP3-NLRP3 interaction and then block its activation. Future studies need to identify the residues of NLRP3 NACHT domain that are responsible for TR binding and clarify the detailed mechanism of how TR blocks NLRP3-NLRP3 interaction by using biochemical and structural approaches."

6. *The Discussion is short and should be expanded*

Reply: Thanks very much for the suggestion and the discussion has been expanded in the revised manuscript.

7. *English needs polishing.*

Reply: Thanks very much, we have modified the text in the revised manuscript.

2nd Editorial Decision

6 February 2018

Thank you for the submission of your revised manuscript to EMBO Molecular Medicine. We have now received the enclosed reports from the referees that were asked to re-assess it. As you will see the reviewers are now globally supportive and I am pleased to inform you that we will be able to accept your manuscript pending a few final amendments.

***** Reviewer's comments *****

Referee #1 (Comments on Novelty/Model System for Author):

This work is of high academic and technical quality, and the findings are novel.

Referee #1 (Remarks for Author):

The authors have addressed my only major point as requested. Although the comparison, did not come out in favour of Tranilast, and think the total data package is now so strong that the work deserves to be presented for the scientific audience.

Referee #2 (Remarks for Author):

The referees' comments have been addressed in a satisfactory manner. I therefore recommend acceptance for publication

Corresponding Author Names: Rongbin Zhou and Wei Jiang

Journal Submitted to: EMBO Mol Med

Manuscript Number: EMM-2017-08689